# An Overview of the Current State of Cell Viability Assessment Methods Using OECD Classification

**DOI:** 10.3390/ijms26010220

**Published:** 2024-12-30

**Authors:** Eneko Madorran, Miha Ambrož, Jure Knez, Monika Sobočan

**Affiliations:** 1Faculty of Medicine, Institute of Anatomy, Histology and Embryology, University of Maribor, Taborska ulica 8, 2000 Maribor, Slovenia; 2Faculty of Medicine, Institute of Translational and Clinical Research, University of Maribor, Taborska ulica 8, 2000 Maribor, Slovenia; miha.ambroz@student.um.si (M.A.); monika.sobocan@gmail.com (M.S.); 3Department for Gynaecologic Oncology and Oncology of the Breast, University Division for Gynaecology and Perinatology, Ljubljanska ulica 5, 2000 Maribor, Slovenia; jure.knez1@um.si; 4Division of Gynaecology and Perinatology, University Medical Centre Maribor, Ljubljanska ulica 5, 2000 Maribor, Slovenia

**Keywords:** cell viability, cell-based methods, in vitro toxicology, OECD cell viability classification

## Abstract

Over the past century, numerous methods for assessing cell viability have been developed, and there are many different ways to categorize these methods accordingly. We have chosen to use the Organisation for Economic Co-operation and Development (OECD) classification due to its regulatory importance. The OECD categorizes these methods into four groups: non-invasive cell structure damage, invasive cell structure damage, cell growth, and cellular metabolism. Despite the variety of cell viability methods available, they can all be categorized within these four groups, except for two novel methods based on the cell membrane potential, which we added to the list. Each method operates on different principles and has its own advantages and disadvantages, making it essential for researchers to choose the method that best fits their experimental design. This review aims to assist researchers in making this decision by describing these methods regarding their potential use and providing direct references to the cell viability assessment methods. Additionally, we use the OECD classification to facilitate potential regulatory use and to highlight the need for adding a new category to their list.

## 1. Introduction

Cell viability, defined as the proportion of living, healthy cells within a given population [1], is crucial in various fields. In the pharmaceutical industry, it plays a pivotal role in screening potential therapeutic agents and determining their safe dosage ranges [2]. In toxicology, cell viability assays help assess the safety of chemicals and environmental pollutants, ensuring compliance with regulatory standards [2]. In cancer research, these assays are essential for testing the efficacy of anti-cancer drugs and developing effective treatment strategies [3]. Ultimately, cell viability tests advance our understanding of cellular processes and the biological effects of compounds [4].

At first, measuring cell viability may sound trivial, but it is difficult to determine whether a cell is alive or dead [5]. A cell is considered viable if it can perform its essential functions, which are many [6]. On the contrary, the NCCD considers a cell to be dead when the plasma membrane’s barrier function is irreversibly lost, the cell forms apoptotic bodies, or the cell is engulfed by professional phagocytes [7]. Considering the many different aspects that define a cell to be dead (or not alive, as both are exhaustive terms), there are many different methods to test cell viability [8]. Some methods measure metabolic activity, which is crucial for a cell’s essential functions, such as the MTT assay and ATP assay [9]. Additionally, some methods focus on cell structure integrity, particularly cell membrane integrity, using indicators like propidium iodide and acridine orange [9]. Other methods assess cell growth or proliferation, as a dividing cell is considered viable, using assays like the proliferation assay and BrdU [9]. It should be noted that cells can be viable without proliferating, whereas proliferating cells are always viable [7]. There are also damaged cells that arrest cell proliferation and secrete inflammatory cytokines and other factors, determined as senescence cells [10]. To organize these methods, the Organisation for Economic Co-operation and Development (OECD) compiled a classification, which we will follow in this manuscript [9]. The OECD provides standardized guidelines and protocols that ensure consistency, reliability, and regulatory compliance in scientific research. These standardized methods are widely accepted and used globally, which helps in comparing and validating results across different studies [11].

Researchers should select a method based on their specific endpoint, available resources (instrumentation, cost, or necessary skills), and the characteristics of the method. This paper aims to provide a comprehensive overview of cell viability assays based on the OECD classification, highlighting their principles, applications, and comparative advantages. By understanding the strengths and limitations of each method, researchers can choose the most appropriate assay for their experimental needs.

## 2. Cell Viability Methods

We will use the OECD list to categorize the various cell viability methods, briefly describe them, and summarize their endpoints, advantages, and disadvantages [9].

### 2.1. Structural Cell Damage (Non-Invasive)

Cell injury that causes changes in cell morphology encompasses the structural and functional changes caused by harmful stimuli or stressors, which can be reversible (e.g., cellular swelling, steatosis) or irreversible (e.g., apoptosis, necrosis), and can be observed through microscopy and biochemical techniques (Figure 1a) [12,13,14,15]. Concretely, the overall shape and size of the cells may be changed due to the compound’s toxic effect on the cell’s cytoskeleton or cell membrane channels [16,17]. Various programs, such as ImageJ 1.54m /Fiji 2.9.0, CellProfiler 4.2.1, Icy 2.1.4, or IDEAS 6.2, can aid in the quantitative determination of cell shape changes [18,19]. However, several factors can induce changes in cell morphology that are not related to the toxic effects of the compounds, such as cell attachment, osmolarity, and metabolism [20,21]. Thus, it is challenging to determine whether the changes in cell morphology are due to the toxic effects of the test compound.

There are other non-invasive tests used to determine the structural cell damage based on markers that leak out of the cytoplasm of dead cells into the cell culture medium [8]. In this category, researchers may measure the release of compounds into the supernatant due to cell membrane rupture (Figure 1b) [8,22]. The most common compound used in this type of method is lactate dehydrogenase (LDH) [8]. However, other compounds are also used under different trademarks, such as adenylate kinase (AK) (ToxiLight™, LONZA, Basel, Switzerland), dead-cell protease (not specify, CytoTox™, Promega, Madison, WI, USA), Glyceraldehyde-3 Phosphate Dehydrogenase (G3PDH) (aCella™—TOX, Promega, Madison, WI, USA), and glucose 6-phosphate dehydrogenase (G6PD) (Vybrant™ Cytotoxicity Assay Kit or CyQUANT™ (Thermo Fisher Scientific, Waltham, MA, USA). These compounds are cytoplasmic enzymes that are present in all cells and are rapidly released when the plasma membrane is damaged [22]. Since permanent membrane disruption is the definitive act of cell death [7], there is a correlation between the amount of these compounds in cell supernatant and cell viability. [22]. However, these compounds can also be released by intactive cell membranes, evidenced by their leakage under stress or metabolism change conditions [23,24]. Furthermore, high background levels of these compounds, specially LDH, have been observed in untreated samples [25,26]. Interestingly, a study found that the LDH method underestimates cytotoxicity when assessing cell viability in co-culture with bacteria [27]. Farhana and Lappin [28] reported issues with long-term assays involving medium changes. Azqueta et al. highlighted LDH harmonization efforts in nanotoxicology due to significant variations in cytotoxicity assays across different laboratories [29]). LDH, WST-1, and MTS assays demonstrated substantial variation in maximal values of cytotoxicity or concentration–response relationships in cell cultures after exposure to nanomaterials [30,31,32,33,34].

### 2.2. Structural Cell Damage (Invasive)

This cell viability category also relies on membrane integrity, but here, the molecules enter non-viable cells (inward direction) [8].

One of the oldest methods to determine cell viability with this characteristic is the trypan blue method [35]. Trypan blue selectively penetrates dead cells with damaged plasma membranes and is impermeable to viable cells (Figure 2a). The selective staining mechanism is related to the impermeability of trypan blue aggregates [36]. Prolonged incubation can result in viable cell staining due to the dissociation of trypan blue aggregates [36]. To address this, incubation periods with trypan blue are kept short, which might lead to the underestimation of dead cells [37]. Many optimized methods exist to minimize these events [37]. The method is cost-effective and various benchtop instruments are available to measure cell viability (hemocytometer, Bio Rad TC10/TC20 Automated Cell Counter; Olympus Cell Counter model R1; ThermoFisher Scientific Countess II Automated Cell Counter (fluorescent); Roche Cedex HiRes Analyzer; Nexcelom Bioscience Cellometer Auto T4) [36].

Aside from trypan blue dye, there are several other dyes that are only transported inside the non-viable cells. Once inside, they bind to various target molecules. Upon binding, these dyes emit fluorescence when excited with a specific wavelength [38]. Each dye has a unique composition, and there are many different types within this category, including propidium iodide, Hoechst 33342, DRAQ7, and acridine orange [39], and some proprietary dyes: CellTox (Promega), SYTOX, YO-YO, and TO-PRO-3 Iodide from Thermo-Fisher Scientific [36]. Numerous studies have validated the correlation between the presence of these dyes and cell viability [13,36,38,39]. However, there are also reports of false-positive events when using these dyes to assess cell viability [40,41,42,43,44]. These false positives can occur due to changes in osmolarity, metabolism, or spontaneous invagination, which may cause the dye to penetrate otherwise viable cells [45].

Within this category, there are lipid-soluble dyes that may cross the cell membrane due to their hydrophobic nature. These dyes are transformed by cellular enzymes (esterases) into lipid-insoluble fluorescent compounds that cannot escape from cells with intact membranes (Figure 2b) [46,47]. The presence of these modified dyes, cleaved by the cellular enzymes, indicates viable cells. However, similar to how LDH can leak from viable cells, these transformed dyes may also leak through viable cells, more so considering the higher cell membrane permeability of cell membranes to lipid soluble molecules [48]. Moreover, the enzymes that cleaved the dyes during the cell death pathway were actually synthesized when the cell was still viable [49].

Numerous tests combine the aforementioned types of dyes—those that cross the membrane when compromised and lipid soluble dyes cleaved by esterases—to achieve better correlation. An example of such a test is the Life/dead kit (Thermo-Fisher Scientific).

Within this category, we also consider the antibodies that attach to molecules related to the activation of cell-death-associated pathways (Figure 2c) [50,51]. In apoptotic pathways, two main events are targeted: caspase activation (detectable through enzymatic analysis or cell staining) [51] and endonuclease activation (observable as DNA fragmentation at the population level) [52]. Additionally, chromatin condensation (visible via DNA staining) and the externalization of phosphatidylserine on the plasma membrane (detectable using annexin staining) strongly correlate with apoptotic cell death [53]. In any case, the target molecules should be associated with processes occurring after the point of no return, as an apoptotic cell may reverse its pathway and remain viable through a process known as anastasis [54]. However, within cell population, there may be sub-populations following different cell death pathways [55], which might remain undetected when using a single endpoint. Moreover, the pathways are interconnected, with crosstalk between different mechanisms complicating the selection of specific markers [56].

### 2.3. Cell Growth

There are many ways of counting cells from a given sample, including optical microscope, flow cytometer, haematocytometer [57], which may also be used to assess cell growth and, subsequently, determine cytotoxicity. By observing the difference in cell numbers before and after treatment with a test compound, we can identify the number of cells that lysed due to the treatment (Figure 3a). Subsequently, by comparing the difference in cell numbers between the treated and untreated samples, we can assess the toxicity of the test compound [58].

This method measures new DNA synthesis based on the incorporation of the easily detectable nucleoside analogues such as Bromodeoxyuridine (BrdU) or EdU into DNA [59]. BrdU is an analogue of thymidine, a nucleoside involved in DNA synthesis. During the S-phase of the cell cycle, BrdU is incorporated into replicating DNA (Figure 3b). BrdU or EdU can be detected, for example, by using fluorescent-labelled antibodies in permeabilized cells [59]. These methods provide an accurate count of the number of cells that have divided because the compound intercalates in the DNA at a known rate [59]. However, BrdU and Edu are cytotoxic and temper with the proliferation ability of the cells, leading to unrealistic results [60,61]. The BrdU protocol is also more time-consuming than the EdU protocol, which does not require a DNA denaturation step [59].

Cell growth may also be measured with dyes that bind to cellular proteins or nucleic acids within cells, allowing for the quantification of cell density and protein content. Among the most common cytotoxicity tests within this category are Sulforhodamine B (SRB) and crystal violet (CV) [62,63]. When using crystal violet, the dye attaches to the proteins that anchor the cell. If the cell detaches (due to cell death), the amount of crystal violet decreases. This makes it useful for measuring attached cells [4]. Anyhow, protein staining serves as a proxy for actual cell count, so these methods measure the total protein content in the cell rather than individual cells (Figure 3c) [62,63], which may result in lower accuracy. Moreover, the use of crystal violet should be limited because its classification as a toxic and CMR (carcinogenic, mutagenic, or toxic for reproduction) chemical [64].

### 2.4. Cellular Metabolism

The use of dyes metabolized by the cell during its biochemical activities is a widely accepted method for cell viability assessment. Typically, mitochondrial metabolism concretely targeted the production of reducing equivalents such as NAD(P)H, which led to the reduction in the tetrazolium dye in viable cells (Figure 4a). The resulting formazan is extracted and measured spectrophotometrically. The rate of formazan formation reflects the activity of essential cellular processes such as respiration [25,65]. These methods are high-throughput, easy, and low-cost, and have a high sensitivity. They are also versatile and can be used for tissue constructs. Many studies have utilized these methods, and they are included in several ISO standards and OECD test guidelines [65,66,67]. However, this method measures the number of viable cells indirectly, which has some known drawbacks [68]. For instance, cells with reduced mitochondrial function may appear non-viable [68]. The test item can inhibit cell metabolism, leading to low assay values that are not necessarily related to cell viability [45]. Furthermore, some test items interfere with the assay by reducing or enhancing the dye, so interference testing is recommended [68]. Polyphenols may interfere with formazan formation [69]; on the contrary, Cisplatin can increase mitochondrial mass, leading to enhanced formazan formation [70]. As discussed with the SRB and CV methods, these measurements are not made in single cells, but rather, they show population-wise metabolism [71]. Some cell cultures require a long time to reduce a sufficient amount of dye, meaning there is no universal time point for defining viability [69]. Assessing the kinetics of dye reduction may be necessary to select the proper incubation time with a tetrazolium dye and to avoid reaching the OD plateau [72].

Resazurin reduction assay (sometimes called alamarBlue™) is similar to tetrazolium reduction assays because it is based on the metabolic activity of the mitochondria. In this assay, resazurin is reduced to resorufin, which has an excitation peak at 571 nm and an emission peak at 584 nm [67]. Due to its similarity with the tetrazolium reduction assay, this method shares similar advantages and drawbacks. The tests have a high sensitivity, can be performed quickly in multi-well dishes, and are relatively low cost [67]. However, cells with reduced mitochondrial function may appear non-viable, and certain endogenous molecules (e.g., superoxide) can interfere with the assay, reducing the dye [73]. Additionally, some cell cultures require a long time to reduce sufficient resazurin, leading to no sharp time point for defining viability.

Other type of dyes directly measure mitochondrial activity rather than molecule reduction (Figure 4b). Specifically, dyes such as JC-1, TMRE, and MitoTracker™ assess mitochondrial membrane potential [74]. Changes in this potential indicate cell viability loss. Quantification is performed using High Content Imaging (HCI) or Fluorescence-Activated Cell Sorting (FACS) [74]. These methods provide single cell information, and are also fast, inexpensive, and high-throughput [74]. Nevertheless, they are not as sensitive as the methods described above. There are many artefacts that affect plasma membrane potential due to changes in shape and the clustering of mitochondria [75]. Moreover, these dyes are prone to bleaching, quenching, and unquenching. And, especially, JC-1 is toxic to the mitochondria [76].

Neutral red is used in a cell organelle function assay to assess lysosomal function. Active cells accumulate this red dye in their lysosomes, while dying cells do not (Figure 4c). The dye incorporation is measured through spectrophotometric analysis [77]. This method is low-cost and is utilized in various ISO standards and OECD test guidelines with a substantial historical database [78]. It typically provides information at the population level and is not suitable for tissue constructs or certain cell lines, especially for testing items that affect lysosome function [30,79]. Quantitative measurements require normalization, such as with protein content [78].

Dying cells fail to produce ATP, consume more ATP, and may lose ATP through plasma membrane perforations (Figure 4d). By comparing the ATP levels of known viable cells, we can assess the viability of test cells. This involves preparing cell lysates and measuring the total ATP content using a luminometric assay [31]. This rapid high-throughput test is non-invasive and keeps the cells intact, allowing for extended monitoring periods. Similar to other metabolism-based cell viability methods, ATP measurements can be inaccurate when there is a reduction or increase in metabolism, as they may overestimate or underestimate cell viability, respectively.

### 2.5. Membrane Potential

Despite the constant upgrade of existing cell viability methods and the designing of novel cell viability methods, these methods fall under the categories determined by the OECD. However, this category takes a different approach and falls outside the scope of the OECD list. It aims to overcome the major drawbacks of the existing cell viability methods and the use of indirect measurements to determine cell viability. These methods focus on directly measuring cell viability following the cell death definition established by the NCCD in 2015, which considers a cell dead when its membrane permanently loses integrity [7]. In this sense, both methods measure the cell membrane potential to assess membrane integrity (Figure 5).

#### 2.5.1. The Membrane Potential Cell Viability Assay (MPCVA)

The Membrane Potential Cell Viability Assay (MPCVA) determines that a cell is dead when its cell membrane depolarizes permanently. This method uses a fluorescence dye that anchors to the cell membrane, changing its intensity based on the membrane potential. Simultaneously, a cell-membrane-permeable DNA stain is added to measure the DNA content. This step improves the accuracy of cell viability assessment as a doubling cell remains depolarized for extended periods [45]. Since the MPCVA is based on a direct approach to determine cell viability, it is less influenced by artifacts compared to methods using indirect measurements [45]. The method is also compatible with live imaging because the fluorescence dye that anchors to the cell membrane is not toxic and does not influence the courses of the cell [45]. In this case, DNA staining is unnecessary as doubling cells are easily identified by simple observation (two daughter cells forming from a parent cell). Anyhow, the method is new and it has not been extensively tested in various scenarios where unpredicted artifacts might occur [45].

#### 2.5.2. The DD Cell-Tox Method

The DD Cell-Tox method is also based on cell membrane potential, but it adopts a more dynamic approach [32]. Additionally, this method takes a holistic view by considering the various outcomes cells experience after exposing them to a toxic agent [32]. The cell number in a population is measured before and after exposure to the toxic agents. Following exposure, the number of dead cells and doubling cells is determined using a membrane-potential-sensing dye and a DNA stain [32]. By combining these three elements—cell number before and after exposure and the number of doubling cells—a prediction of cell population dynamics is made [32]. This method more accurately assesses the toxic effect of the compounds because it considers the various outcomes triggered by the toxic compounds [32]. It also considers cell population dynamics, providing valuable information about the effect of the toxic agent on the cell population [32]. However, the method is not compatible with live imaging unless a DNA dye is used that does not influence cell processes. The method has not been yet tested in different alternative situations; thus, there are still many unknowns to be addressed [32].

Below are two tables. Table 1 provides a brief description of the method categories, highlighting the advantages and disadvantages of each category. Table 2 lists specific assays to each method category, along with corresponding references for detailed protocol descriptions.

## 3. Discussion

This manuscript presented many different cell viability methods, along with their respective advantages and disadvantages. Ultimately, it is up to the researcher to determine which method best suits better their experimental design (Figure 6).

If the experimental design requires monitoring the cells, a non-invasive method should be chosen because any alteration of the cell itself or its processes will influence the outcome of the study [104]. However, alternative methods should be chosen if high sensitivity is required due to the lower sensitivity of these methods [25,26], (Figure 3a).

A metabolism-based method may be used if the experimental design does not require monitoring and the pathway triggered by the toxic compound is unknown. These methods are not specific to any cell death pathway and are reliable [67] (Figure 3b).

If the experimental design foresees metabolism changes, another approach should be chosen. Methods that rely on cell membrane integrity, such as the use of dyes that crossed the cell membrane when it is allegedly disrupted (e.g., propidium iodide) are commonly used [9] (Figure 3c). However, many studies have observed that these dyes can cross intact cell membranes [40,41,42,43,44]. To avoid such issues, these dyes are often combined with dyes that are esterified within the cell, despite the fact that they, too, have drawbacks [105]. It is worth noting that the esterases responsible for cleaving these dyes may have been synthesized when the cells were viable, even if the cells are currently undergoing a cell death pathway but still have active esterases [106].

Under similar conditions (unknown cell death pathway), the BrdU/EdU methods provide the exact result about the number of cells that divided, as these dyes are intercalated in the DNA [59]. However, they are toxic and may influence the cell viability [60,61]. Similarly, fluorescence dyes that bind to cellular proteins follow the same principle, but they are less accurate and have similar drawbacks [64].

If the pathway is known, a more specific method using antibodies may be selected to gain a more comprehensive understanding of the expected cell death pathway [50,51]. However, this method does not encompass alternative cell death pathways that may occur within the same cell population [55]. Thus, combining it with a general cytotoxicity test is necessary, since antibody-based techniques may unintentionally select specific subpopulations and overestimate cell viability [50].

In this context, a more universal method may aid researchers evaluating the viability of all subpopulations within a cell population. Viability methods based on the cell membrane potential [5,32,45] are less prone to artifacts, since they directly assess cell death [32,45]. The MPCVA has the advantage of being compatible with live imaging, while the DD Cell-Tox method is of special interest because it encompasses all possible outcomes triggered by the tested molecule [32,45]. This is particularly valuable in cancer research for testing anti-carcinogenic drug biological effects in the cell [32]. However, both methods are new, and their accuracy and reliability need to be evaluated in various alternative situations.

It is also worth mentioning that new technologies enhance cell viability assessment methods. For instance, benchtop fluorescence microscopes with fluorescence emission capabilities can be measured using various instruments such as EVOS, Axio Observer.Z1, or Opera Phenix [107]. Similarly, new spectrophotometers, like the Varioskan LUX, can measure luminescence, making this measurement more accessible. Nowadays, instruments capable of live imaging are more readily available, making them powerful tools for cell viability studies (Cytation, Synergy Neo 2, TriStar2S LB 942) [108]. Of particular interest is the multispectral imaging flow cytometer, known as ImageStream MK2, which combines flow cytometry and fluorescence microscopy techniques [19].

Therefore, there are numerous methods and instruments to determine cell viability, as well as many studies comparing different cell viability methods [1,109,110]. Moreover, researchers can assess the accuracy of each test by challenging cells with the standard panel of toxicants. However, few studies challenge the accuracy of the methods against a standard method [109]. The ideal method should adopt a single-cell approach and consider that most cell death pathways last up to 48 h with minimal false positives and false negatives [45]. It should also consider that some cell death pathways involve senescent cells, whose cell death pathway can last for days. This process is not yet fully understood, but the cell response through accelerated senescence includes the induction of reversible polyploidy and the possibility of reprogramming towards self-renewal, particularly in cancer [111]. Thus, cells should be stained with a DNA dye to observe any cell cycle arrest 48 h after the induction, which will likely indicate the senescence pathway. Counting cells 48 h after their exposure to a toxic agent fulfils all these requirements and was successfully tested when testing the MPCVA method’s accuracy [45]. Even though this method is time-consuming, it is a strong candidate for a standard method to test the accuracy of less time-consuming cell viability methods. If the tested methods provide comparable viability assessment, they can be further used to determine cell viability under the same conditions. The OECD should adopt this approach to ensure scientific soundness, reliability, and regulatory compliance for novel cell viability methods.

## 4. Conclusions

Over the past 100 years, numerous cell viability methods have been developed and some have been updated in recent years. New methods have also been developed by leveraging recent advancement in knowledge and developed technology. Consequently, there are numerous cell viability methods based on different principles that are suitable for various experimental designs, and researchers can select the most suitable method for their experimental designs. This manuscript aims to assist researchers selecting the cell viability method that better suites their experimental design needs. This manuscript utilizes the OECD categorization to facilitate a smoother transition for authors who may intend to use this method for regulatory purposes. Additionally, we introduce new cell viability methods that do not fit within the existing categories of the OECD list. To accommodate these methods, we added a new category: cell-membrane-potential-based methods. Additionally, we highlighted the lack of information regarding accuracy assessment comparisons between different methods and suggested the cell count method as a benchmark for comparison by the OECD.

## 5. Future Directions

Researchers should develop new cell viability methods based on innovative approaches that directly measure cell viability and seek their incorporation into the OECD list. These new methods should be rigorously tested against a highly accurate standard cell viability method.

## Figures and Tables

**Figure 1 ijms-26-00220-f001:**
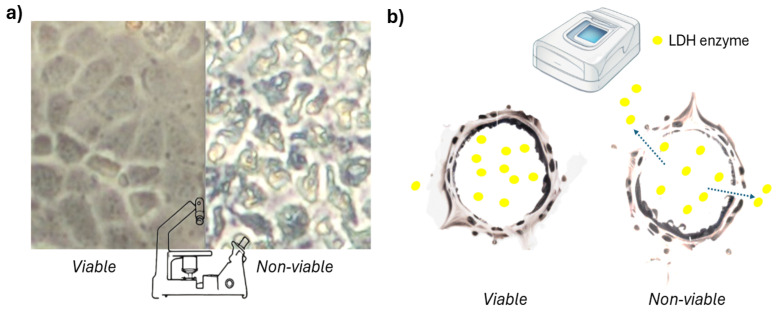
Graphical scheme of the cell viability methods based on structural cell damage (non-invasive). (**a**) Cell morphology changes between two samples with an inverted microscope. The left image shows viable cells, while the right image displays cells treated with 12.5 mg/L of Arsenic V. (**b**) Measurement of leaking molecules from viable and non-viable cells. The provided example uses the LDH enzyme, which leaks in greater amounts from non-viable cells due to membrane disruption. The supernatant (extracellular fluid) is collected and measured with a spectrophotometer, where higher values indicate non-viable cells.

**Figure 2 ijms-26-00220-f002:**
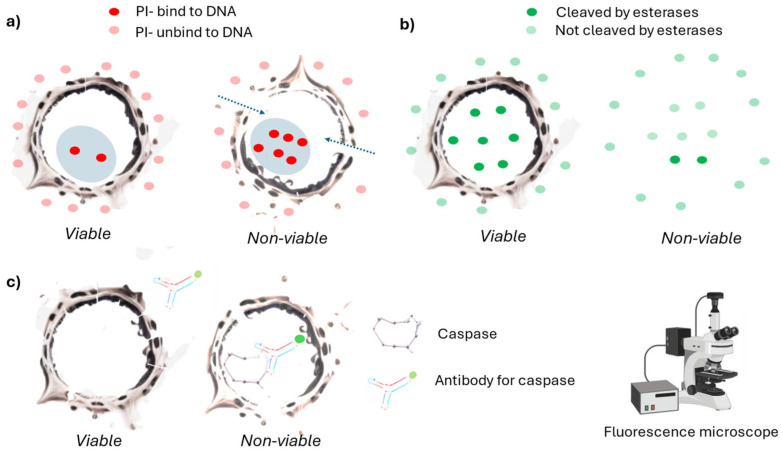
Graphical scheme of the cell viability methods based on structural cell damage (invasive). (**a**) In non-viable cells, a molecule like PI is transported within the cell. While it can also be transported in viable cells, it occurs to a much lesser extent. Once PI binds to RNA or DNA, it emits fluorescence upon excitation. (**b**). Lipophilic dyes pass through the cell membrane and are cleaved by esterases within viable cells, occurring more frequently in viable cells. After cleavage, they emit fluorescence upon excitation. (**c**) Antibodies pass through permeabilized cells and attach to specific molecules related to cell death pathways, such as caspase. If the target molecule is present, the antibodies bind to them, allowing visualization or quantification using fluorescence measurement instruments.

**Figure 3 ijms-26-00220-f003:**
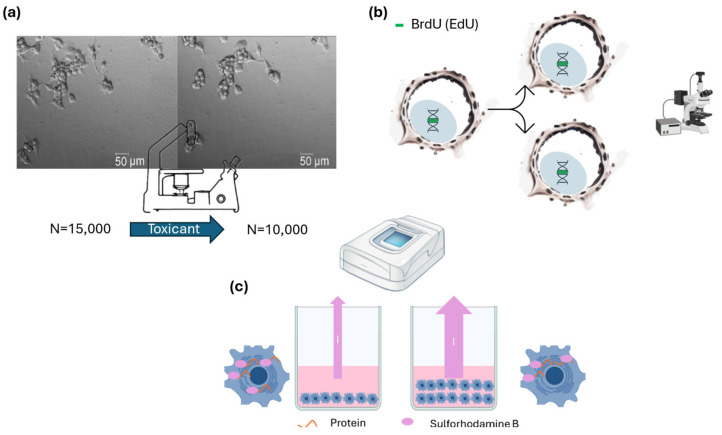
Graphical scheme of the cell viability methods based on cell growth. (**a**) The cell count of a sample is assessed before and after treatment using an inverted microscope or, alternatively, a flow cytometer. (**b**) BrdU or EdU are integrated into the cell DNA, making them visible in daughter cells and allowing for the precise quantification of replicated cells. (**c**) The absorbance of Sulforhodamine B, which binds to cellular proteins, can be measured with a spectrophotometer. Thus, this measurement enables the quantification of the total cell protein content.

**Figure 4 ijms-26-00220-f004:**
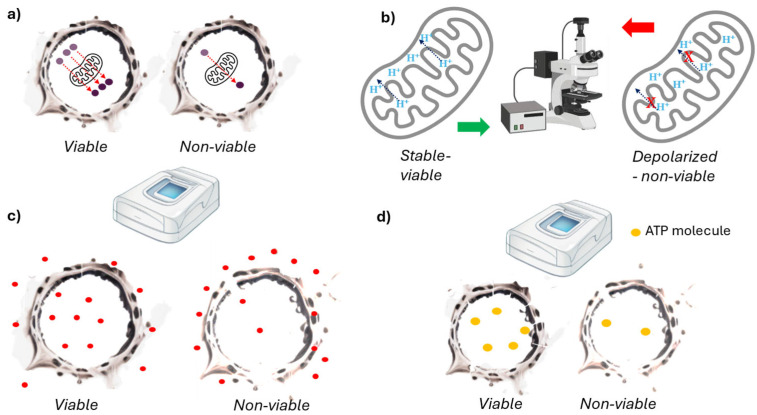
Graphical scheme of the cell viability methods based on cellular metabolism. (**a**) Molecules are metabolized in the mitochondria at varying rates depending on the cell’s metabolism (viability). The viable cell on the left has a higher number of reduced molecules compared to the non-viable cell on the right, resulting in lower absorbance or fluorescence intensity in the latter. (**b**) The mitochondrial membrane potential is measured using a dye that emits different intensities and bandwidths based on the mitochondrial membrane potential. (**c**) Neutral red is transported within the cell by lysosomes. In viable cells, this transport is unaltered, while in non-viable cells, it is disrupted. The amount of neutral red within the cell is then measured, with higher levels found in viable cells. (**d**) Viable cells produce more ATP than non-viable cells. After cell lysis, ATP content is measured using luminometric assays.

**Figure 5 ijms-26-00220-f005:**
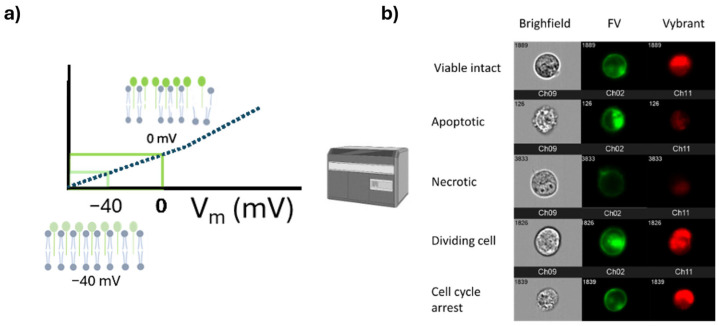
Graphical scheme of the cell viability methods based on cell membrane potential. (**a**) The intensity of dye fluorescence is proportional to the membrane potential. Previous studies utilized Fluovolt™ (FV) (Thermo Fisher Scientific, Waltham, MA, USA) for this purpose. (**b**) Cell viability is assessed by measuring both the cell membrane potential and DNA content using the dye Vybrant™. Viable cells and cells in cell cycle arrest exhibit stable FV intensity, while apoptotic and dividing cells show higher intensity. Necrotic cells have the lowest fluorescence emission due to dye leakage. Dividing cells and cells in cycle arrest emit higher fluorescence intensity corresponding to 2n DNA.

**Figure 6 ijms-26-00220-f006:**
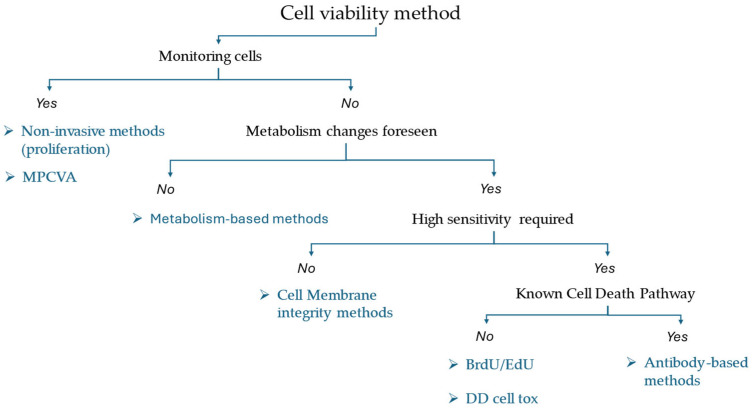
Suggested decision tree for the desired method.

**Table 1 ijms-26-00220-t001:** Comparative analysis of cell viability methods.

Method (Group)	Principle	Endpoint (Instrument)	Advantages	Disadvantages
Structural cell damage (non-invasive)				
Optical microscope (1)	Changes in the morphology of the cell.	Observed changes in the morphology of the cell (optical microscope).	Quick, cheap, and it could be automatized.	Many artifacts affect the accuracy of the assay.
Release of intracellular compounds (2)	Measures the release of compounds into the cell culture medium due to damaged cell membrane.	Absorbance or luminescence of the compound (spectrophotometer or luminometer).	Quick, non-invasive.	High background level and potential false-positive events.
Structural cell damage				
Trypan blue (3)	Viable cells exclude the trypan blue dye.	Observation of stained cells (optical microscope).	Cheap, it could be automatized.	High rates of false positives and false negatives.
Lipid-soluble dyes (4)	Hydrolysis of the dye by intracellular esterase.	Fluorescence intensity of reduced products (fluorescence microscope, flow cytometer).	Quick, versatile (analysis of single cell or cell population in different instruments).	High rate of false positives.
Propidium iodide (5)	The dyes are impermeant only to live cells.	Fluorescence intensity of the internalized dye (fluorescence microscope, flow cytometer).	Quick, many references.	High rate of false positives.
Live/Dead assay (6)	It has two dyes: one impermeant only to live cells and the second is cleaved by esterases within the cell.	Fluorescence intensity of internalized dye and reduced dye (fluorescence microscope, flow cytometer).	Quick, versatile, better accuracy than single dyes alone.	High rate of false positives.
Antibodies to cell-death-associated pathways (7)	Determines the presence of molecules associated to cell death pathways.	Fluorescence intensity of the dye attached to the antibody (fluorescence microscope, flow cytometer).	Specific, determines the type of the programed cell death.	Expensive, unable to determine the viability of the cells with a different cell death pathway.
Cell growth				
Cell division (8)	Difference in cell number before and after the exposure.	Counting cell number in a population (microscope, flow cytometer, cell counter).	Cheap, accurate, and straightforward.	Time-consuming.
BrdU/EdU (9)	Daughter cells contained BrdU intercalated in their DNA.	Fluorescence intensity of BrdU (fluorescence microscope, flow cytometer).	High sensitivity no.	Toxic, impairs cell division.
Sulforhodamine B (SRB) and crystal violet (10)	The amount of dye is proportional to the cells (cell proliferation).	Absorbance of the dye (spectrophotometer).	Quick, cheap.	Toxic, measures cell mass, not cell viability.
Cellular metabolism				
MTT (11)	Reduction in the tetrazolium dye to formazan.	Absorbance of the formazan product (spectrophotometer).	Quick, cheap, high throughput, and many references.	Many artifacts affect the accuracy of the assay.
alamarBlue™ (12)	Resazurin reduction to resorufin.	Luminescence measurement of resorufin (fluorescence microscope and flow cytometer).	Quick, cheap, and high throughput.	Naturally occurring molecules disrupt the assay.
JC-1, TMRE, MitoTracker™ (13)	Mitochondrial membrane potential (lost in non-viable cells).	Fluorescence emission of the dye proportional to the mitochondrial membrane potential (fluorescence microscope and flow cytometer).	Fast, cheap, and high throughput.	Prone to bleaching, quenching, and unquenching. Accuracy issues in certain situations.
Neutral red (14)	Viable cells incorporate and bind the neutral red dye.	Absorbance of the incorporated dye (spectrophotometer).	Quick, cheap, and standardize.	Not a good correlation. Lysosomal activity affects its accuracy.
ATP production (15)	ATP production is correlated to cell viability.	Luminescence measurement of released ATP (luminometer).	Live imaging, non-invasive, and high throughput.	Many artifacts affect the accuracy of the assay.
Cell membrane potential				
MPCVA (16)	The cell membrane potential determines cell membrane integrity.	Fluorescence intensity of the dyes (fluorescence microscope, flow cytometer).	Direct determination, live imaging.	Not tested in various alternative situations.
DD Cell-Tox (17)	The cell membrane potential determines cell membrane integrity and DNA content in the doubling cells.	Fluorescence intensity of the dyes (fluorescence microscope, flow cytometer).	Direct determination, considers the cell population dynamics and the various outcomes triggered by toxic compounds.	Not tested in various alternative situations. Not suitable for long periods of cultivation.

**Table 2 ijms-26-00220-t002:** Cell viability method list.

Group	Method	References
1	Label-free imaging	[15]
2	LDH (Promega, Madison, WI, USA)	[23]
2	CytoTox^TM^ (Promega, Madison, WI, USA)	[33]
2	Toxi-Light^®^ (LONZA, Basel, Switzerland)	[33]
2	aCella™—TOX (Promega, Madison, WI, USA)	[33]
2	CyQUANT™ (Thermo Fisher Scientific, Waltham, MA, USA)	[34]
3	Trypan blue (MercK KGaA, Darmstadt, Germany)	[80]
4	CellTrace™ prolifereation kit (Thermo Fisher Scientific, Waltham, MA, USA)	[81]
4	Propidium iodide (MercK KGaA, Darmstadt, Germany)	[53]
4	Hoechst 33342 (Thermo Fisher Scientific, Waltham, MA, USA)	[82,83]
4	DRAQ7 (BD Biosciences, San Jose, CA, USA)	[84]
4	Acridine orange (Thermo Fisher Scientific, Waltham, MA, USA)	[85]
4	CellTox (Promega, Madison, WI, USA)	[86]
4	SYTOX (Thermo Fisher Scientific, Waltham, MA, USA)	[87]
4	YO-YO (Thermo Fisher Scientific, Waltham, MA, USA)	[88]
4	TO-PRO-3 Iodide (Thermo Fisher Scientific, Waltham, MA, USA)	[89]
5	Calcein AM (Thermo Fisher Scientific, Waltham, MA, USA)	[90]
5	CytoCalcein™ (Thermo Fisher Scientific, Waltham, MA, USA)	[91]
5	Propidium iodide (MercK KGaA, Darmstadt, Germany)	[92]
6	Live/Dead assay (Thermo Fisher Scientific, Waltham, MA, USA)	[93]
7	Antibodies	[94]
7	ApoTox-Glo (Promega, Madison, WI, USA)	[95]
7	Annexin V (Thermo Fisher Scientific, Waltham, MA, USA)	[96]
8	Cell division counting	[58]
9	BrdU assay (MercK KGaA, Darmstadt, Germany)	[59]
9	Edu assay (MercK KGaA, Darmstadt, Germany)	[59]
10	Sulforhodamine B (MercK KGaA, Darmstadt, Germany)	[97]
10	Crystal Violet (MercK KGaA, Darmstadt, Germany)	[98]
11	MTT assay (MercK KGaA, Darmstadt, Germany)	[65]
11	WST-1 assay (MercK KGaA, Darmstadt, Germany)	[99]
11	MTS assay (MercK KGaA, Darmstadt, Germany)	[8]
11	XTT assay (MercK KGaA, Darmstadt, Germany)	[100]
12	alamarBlue™ (Thermo Fisher Scientific, Waltham, MA, USA)	[101]
13	JC-1 (Thermo Fisher Scientific, Waltham, MA, USA)	[74]
13	TMRE (Thermo Fisher Scientific, Waltham, MA, USA)	[102]
13	MitoTracker™ (Thermo Fisher Scientific, Waltham, MA, USA)	[102]
14	Neutral red (MercK KGaA, Darmstadt, Germany)	[103]
15	ATP assay (Promega, Madison, WI, USA)	[8]
16	MPCVA	[45]
17	DD Cell-Tox	[32]

## Data Availability

Not applicable.

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
