# Peer review of "An Overview of the Current State of Cell Viability Assessment Methods Using OECD Classification"

_ijms, 2024, doi:10.3390/ijms26010220_

Round 1

Reviewer 1 Report

Comments and Suggestions for Authors

ijms-3379838-peer-review

The current state of cell viability methods

This review article compiles the known methods for detecting cell viability in vitro up to the current state. Indeed, the problem of using the appropriate cell vitality detection methods is actual, in particular as concerned with the new methods and most modern instrumentation.

  The authors decided to base it on the full list of toxicology tests developed by the OECD (dated 2016, ref 7). What means this abbreviation is not explained in the first mention, as well as the second, more sophisticated and less used here source of information NCCD (dated from 2015, ref 9). The latter is assuming for decades and replenishing more scientifically unequivocal information on cell death, however, the last NCCD report from 2023 (doi: 10.1038/s41418-023-01153-w) is not cited here.

 More than 20 methods, beginning with the oldest and simplest vital stains, are grouped into five categories. The essence of each group is schematically presented in five figures with very short descriptions in the corresponding sections. All these are summarised in Table 1 with an indication of the assessment instrument, advantages and disadvantages of each method. In Table 2, the references for the protocols are given. That is convenient.

 The authors state in the Introduction, that cell vitality detection methods are mostly used in toxicology and are important in cancer research. For cancer, they refer to the review by two prominent experts Mirzayans and Murray from 2018 (ref 3) but it is seen that they have not read its content at all.

Besides cell death by apoptosis, Roninson et al described in 2001 another response of proliferation interruption by still metabolically active cells, cellular senescence (Roninson et al 2001  doi: 10.1054/drup.2001.0213.), which is not mentioned in the current review. It is quite a consensus of the last ten years (many articles, chapters and even books) to deal with it in the current state of cancer cell biology. This complex, not yet fully understood cell response by accelerated senescence includes induction of reversible polyploidy and the option of reprogramming towards self-renewal, particularly in cancer (Moiseniak, Sikora 2010 Polyploidy: the link between senescence and cancer DOI: 10.2174/138161210790883714). Besides other stresses, the senescence response to cancer therapy, which is undoubtedly toxic, allows resistance to anticancer medicines to take place, just through both senescence and polyploidy reverse (review Erenpreisa et al. 2020, Paradoxes of cancer: survival at the brink doi.org/10.1016/j.semcancer.2020.12.009). For cell viability metabolic function, just this aspect is discussed in ref 3  because such senescent polyploidy cells remain metabolically active and positive in the MTT test for weeks. So, cells may not die in 48 hs as the authors of the current review say to us. This important work by Mirzayans and Murray is not consumed.

 Moreover, regarding apoptosis, more recently, new data from serious laboratories were discussed in several articles on its reverse in some cases, even from late stages (anastasis). By the way, it was shown that activated caspase 3 can mark both apoptosis and anastasis (e.g., Zaitseva et al. 2021 doi: 10.3390/cancers13153671), so the test with caspase 3 alone, as presented by current authors, is no longer a suitable marker of apoptosis.   

 From the same, it follows that cell proliferation detection is not always an appropriate test for cell viability. In the state of accelerated senescence, cells do not divide and do not increase in cell counts, but they do replicate DNA and include BrdU. This was also shown many-fold in our lab.

 The review also suffers from the non-precise title and section subtitles. The review title is “The current state of cell viability assessment methods”.

I judge that cell viability is a cell state with certain functions, while the methods of their detection – represent the assessment of these functions. The present title mixes the causes and reasons up.

Also,  the subtitles: “Cell damage methods (non-invasive and invasive)” should be précised as “Cell membrane damage and leakage (invasive and not invasive methods of assessment”), correspondingly.

 The best part of this review is the description of cell membrane potential methods, to which the authors have a significant contribution and description of the newest live-microscopy methodology.

 In conclusion,

The present review article on cell viability is outdated. To be prepared as due, the review should be divided into two parts: (1) The problem of cell viability in its current updated context and (2) The methods of cell viability assessment, where the second is determined by the update of the first. In its present variant, the knowledge lack of the first leads in some cases to the doubtful recommendations in the second, methodological part. As well, it refers to the discussed accuracy of the methods. It should be adequate for the study task, of course, but also considered that if an experiment is monitoring the reversible and fluctuating processes, such as cell senescence, some special approach (repeated frequent testing) is needed. I judge that such kind of a major revision is hardly rational and feasible.

 Therefore, I cannot recommend this article for publication in IJMS. But this text, without major changes, may be very useful for discussion in some Internet blogs for the exchange of opinions and experience among scholars and less experienced students. Alternatively, it could be literary edited and reduced to just the most modern methodology, which may interest the readers of this journal and where the authors have good expertise to share.

Comments on the Quality of English Language

English is not bad as much as I can judge,

Author Response

The current state of cell viability methods

This review article compiles the known methods for detecting cell viability in vitro up to the current state. Indeed, the problem of using the appropriate cell vitality detection methods is actual, in particular as concerned with the new methods and most modern instrumentation.

The authors decided to base it on the full list of toxicology tests developed by the OECD (dated 2016, ref 7). What means this abbreviation is not explained in the first mention, as well as the second, more sophisticated and less used here source of information NCCD (dated from 2015, ref 9). The latter is assuming for decades and replenishing more scientifically unequivocal information on cell death, however, the last NCCD report from 2023 (doi: 10.1038/s41418-023-01153-w) is not cited here.

We concur with the author that the OECD classification is outdated; however, it remains the current standard and has not been revised. Although including the OECD classification is uncommon, it is of utmost importance because their guidelines ensure standardization and are mandated by regulatory bodies for the approval of new drugs, chemicals, and other products to ensure their safety. This highlights another gap needing investigation: many valuable viability methods are not employed for product approval because regulatory bodies do not require them.

Similarly, the NCCD's current definition of cell death, established in 2015, has not been updated. Despite we find the NCCD reports from 2023 and 2018 very interesting the topics they discuss fall outside the scope of our manuscript, as we do not focus on the various cell death pathways.

More than 20 methods, beginning with the oldest and simplest vital stains, are grouped into five categories. The essence of each group is schematically presented in five figures with very short descriptions in the corresponding sections. All these are summarised in Table 1 with an indication of the assessment instrument, advantages and disadvantages of each method. In Table 2, the references for the protocols are given. That is convenient.

We appreciate the reviewer's comment. As we explain below, our primary target audience for this manuscript is early-stage researchers who need to perform cell viability tests. We believe this manuscript is highly valuable, as there is currently no other publication that provides this essential information.

The authors state in the Introduction, that cell vitality detection methods are mostly used in toxicology and are important in cancer research. For cancer, they refer to the review by two prominent experts Mirzayans and Murray from 2018 (ref 3) but it is seen that they have not read its content at all.

We apologize for the reference mismatch. We recently transitioned from one reference editor to other, and as a result, some references did not correspond to the text. We have corrected this issue in the revised version.

Besides cell death by apoptosis, Roninson et al described in 2001 another response of proliferation interruption by still metabolically active cells, cellular senescence (Roninson et al 2001  doi: 10.1054/drup.2001.0213.), which is not mentioned in the current review. It is quite a consensus of the last ten years (many articles, chapters and even books) to deal with it in the current state of cancer cell biology. This complex, not yet fully understood cell response by accelerated senescence includes induction of reversible polyploidy and the option of reprogramming towards self-renewal, particularly in cancer (Moiseniak, Sikora 2010 Polyploidy: the link between senescence and cancer DOI: 10.2174/138161210790883714). Besides other stresses, the senescence response to cancer therapy, which is undoubtedly toxic, allows resistance to anticancer medicines to take place, just through both senescence and polyploidy reverse (review Erenpreisa et al. 2020, Paradoxes of cancer: survival at the brink doi.org/10.1016/j.semcancer.2020.12.009). For cell viability metabolic function, just this aspect is discussed in ref 3  because such senescent polyploidy cells remain metabolically active and positive in the MTT test for weeks. So, cells may not die in 48 hs as the authors of the current review say to us. This important work by Mirzayans and Murray is not consumed.

We agree with the author that the celullar senescence is very important. Yet, as we explained above, the cell death pathways are out of the scope of the manucript. We focus on cell viability tests, as it will be unfeasible to discuss all the topics related to the cell death (cell viability tests, cell death pathways, cellular sensecence, oncogenes and tumor suppressor genes, epigenetic response, toxins...).

We acknowledge that not all cells die within 48 hours after the induction, sometimes it takes minutees, sometimes even days. But the majoritz of cell death pathways end within 48-hours: Messam and Pittman  (Exp Cell Res., 1998), Green (Cell, 2005), Elmore (Toxicol Pathol., 2007), Lekshmi (Cell Death Discovery, 2017) and Newton (Cell review, 2024). Additionally, the standard protocol for cell viability tests typically involves inducing stress and assessing cell viability after 24 hours.

Moreover, regarding apoptosis, more recently, new data from serious laboratories were discussed in several articles on its reverse in some cases, even from late stages (anastasis). By the way, it was shown that activated caspase 3 can mark both apoptosis and anastasis (e.g., Zaitseva et al. 2021 doi: 10.3390/cancers13153671), so the test with caspase 3 alone, as presented by current authors, is no longer a suitable marker of apoptosis.  

We are unable to locate the reference in the text where we mention that the test with caspase 3 alone is a suitable marker of apoptosis. We would appreciate it if the reviewer could point us to where we can find this reference in the manuscript.

From the same, it follows that cell proliferation detection is not always an appropriate test for cell viability. In the state of accelerated senescence, cells do not divide and do not increase in cell counts, but they do replicate DNA and include BrdU. This was also shown many-fold in our lab.

We concur with the reviewer that cell proliferation tests are not always appropriate for assessing cell viability. The main goal of this manuscript is, precisely, to describe the various tests available for evaluating cell viability and to identify the test that best suits the experimental design. We acknowledge that each test has its own advantages and disadvantages. Its a matter of finding the most suitable test.

The review also suffers from the non-precise title and section subtitles. The review title is “The current state of cell viability assessment methods”.

I judge that cell viability is a cell state with certain functions, while the methods of their detection – represent the assessment of these functions. The present title mixes the causes and reasons up.

We agree with the reviewer that the original title was not suitable for the manuscript, so we have revised it accordingly. We hope the new title better encompasses and describes the manuscript's content.

Also,  the subtitles: “Cell damage methods (non-invasive and invasive)” should be précised as “Cell membrane damage and leakage (invasive and not invasive methods of assessment”), correspondingly.

We have adopted the same terminology as the OECD to structure the various cell viability tests. As mentioned earlier, we include the OECD classification because, while its usage in research is uncommon, it is required for regulatory purposes.

The best part of this review is the description of cell membrane potential methods, to which the authors have a significant contribution and description of the newest live-microscopy methodology.

We are grateful for the reviewer's positive feedback, which inspires us to further develop the method.

In conclusion,

The present review article on cell viability is outdated. To be prepared as due, the review should be divided into two parts: (1) The problem of cell viability in its current updated context and (2) The methods of cell viability assessment, where the second is determined by the update of the first. In its present variant, the knowledge lack of the first leads in some cases to the doubtful recommendations in the second, methodological part. As well, it refers to the discussed accuracy of the methods. It should be adequate for the study task, of course, but also considered that if an experiment is monitoring the reversible and fluctuating processes, such as cell senescence, some special approach (repeated frequent testing) is needed. I judge that such kind of a major revision is hardly rational and feasible.

Therefore, I cannot recommend this article for publication in IJMS. But this text, without major changes, may be very useful for discussion in some Internet blogs for the exchange of opinions and experience among scholars and less experienced students. Alternatively, it could be literary edited and reduced to just the most modern methodology, which may interest the readers of this journal and where the authors have good expertise to share.

We thank the reviewer for their constructive comments. They made us realize that the original title misled readers and shifted their focus away from the intended goals of the manuscript. As the reviewer foresaw, the manuscript is specifically targeted at students and early career researchers. There is a significant demand for a paper that describes viability methods and where to find them. Students often turn to forums or blogs for advice, but when seeking specific protocols, they consult reputable journals like the IJMS. Additionally, the inclusion of the OECD classification is a valuable addition for experienced researchers who want to align their research with regulatory purposes. We are currently involved in the EUnetCCC project, where we aim to suggest the implementation of viability test methods for regulatory purposes. We added another subtitle for cell membrane potential because these methods cannot be categorized within the existing OECD classification. Otherwise, we would adhere strictly to the OECD classification.

Reviewer 2 Report

Comments and Suggestions for Authors

Although the paper topic is valuable, the manuscript is very poorly written. It significantly lacks important infomation for several described tests. It should be restructured entirely, to be sent as an oppinion paper. Major and minor issues are given below:

In the Introduction section there should be more explanations on the differences in measuring viability vs proliferation, and cytotoxicity vs cytostasis.

In the Abstract, rephrase the sentence” In this sense, there is insufficient information regarding their accuracy, highlighting the need for researchers to investigate the accuracy of both existing and future methods.” This is not true. You can assess the accuracy of each test by challenging cells with the standard panel of toxicants.

Introduction line 40, replace” death” with dead.

At the beginning of Section 2.1. there is a following text “With an optical microscope we can see cells and can even see internal cellular structures such as mitochondria (on average about 2 μm by 0.5 μm) [10, 11]. With this in mind, researchers may analyse  the overall cell shape and size of the cells that may be changed due to the toxic effect of compounds and may be an indicator of the cell viability” This part should be omitted as it is not sufficiently precise or informative. Replace with following text” Cell injury that causes changes in cell morphology encompasses the structural and functional changes caused by harmful stimuli or stressors, which can be reversible (e.g., cellular swelling, steatosis) or irreversible (e.g., apoptosis, necrosis), and can be observed through microscopy and biochemical techniques.”

Figure 1 change expression LDH molecule to LDH enzyme.

Line 102 “Azqueta et al. noted that the LDH method is not very sensitive for determining cytotoxicity” Explain in what way it is not sensitive enough.

Line 150 “Moreover, the enzymes that cleaved the dyes when the cell was in a cell death pathway but they were synthesised at the time the cell was viable [44]”is not understandable. Rephrase it.

Section in line numbers 193-200: You disscuss about use of crystal violet as fluorescent dye but it is actually  an alkaline dye that binds to DNA in the nucleus of a cell. It is not fluorescent. Example of fluorescent dye that binds DNA and reflects the number of cell divisions is CFSE which should be mentioned in this part of the text instead of crystal violet.

Line 197 for crysal violet you state “so this method measures the population mass rather than individual cells”. It measures the number of adherent cells.

Line 215” Moreover, some test items interfere with the assay e.g., by reducing the dye because interference testing is recommended [62)” Provide examples of MTT reduction such as reduction in specific cell mediums, or substances such as polyphenols etc.

Lines 264,265 rephrase in one better structured sentence“Rapid and high-throughput test. The test is not invasive, and the cells are kept intact, which enables monitoring the cells for longer periods”

Line 266 How do you mean it measures cell mass?

Line 269 to 271 The statements are not clear. What is the point in this text?

Figure 3 is missing

Line 365 change the term anti-cancerogenic

Line 369-377 This is not a good description of the instruments used nor the approaches they offer for assessing cell viability. Rewrite this section.

Section 377 to 378 is not clear, Rewrite it.

Comments on the Quality of English Language

English should be significantly improved, as the manuscript is written with poor language usage.

Author Response

Although the paper topic is valuable, the manuscript is very poorly written. It significantly lacks important infomation for several described tests. It should be restructured entirely, to be sent as an oppinion paper. Major and minor issues are given below:

In the Introduction section there should be more explanations on the differences in measuring viability vs proliferation, and cytotoxicity vs cytostasis.

We agree with the reviewer that this is valuable information. We have revised the introduction to include more explanations regarding viability versus proliferation, as the OECD classification considers the category of cell growth. However, we did not add additional explanations for the differences between cytotoxicity and cytostasis, as we believe this is outside the scope of the manuscript, despite being an interesting topic.

In the Abstract, rephrase the sentence” In this sense, there is insufficient information regarding their accuracy, highlighting the need for researchers to investigate the accuracy of both existing and future methods.” This is not true. You can assess the accuracy of each test by challenging cells with the standard panel of toxicants.

We apologize for any confusion. Our intention was to emphasize that there are no references comparing the accuracy of these tests, and few manuscripts make such comparisons between different cell viability tests. However, we believe this is not relevant to our manuscript, which is aimed at aiding in the selection of the appropriate method for experimental design.

Introduction line 40, replace” death” with dead.

We have addressed the issue as recommended by the reviewer.

At the beginning of Section 2.1. there is a following text “With an optical microscope we can see cells and can even see internal cellular structures such as mitochondria (on average about 2 μm by 0.5 μm) [10, 11]. With this in mind, researchers may analyse  the overall cell shape and size of the cells that may be changed due to the toxic effect of compounds and may be an indicator of the cell viability” This part should be omitted as it is not sufficiently precise or informative. Replace with following text” Cell injury that causes changes in cell morphology encompasses the structural and functional changes caused by harmful stimuli or stressors, which can be reversible (e.g., cellular swelling, steatosis) or irreversible (e.g., apoptosis, necrosis), and can be observed through microscopy and biochemical techniques.”

We have addressed the issue as recommended by the reviewer.

Figure 1 change expression LDH molecule to LDH enzyme.

We have addressed the issue as recommended by the reviewer.

Line 102 “Azqueta et al. noted that the LDH method is not very sensitive for determining cytotoxicity” Explain in what way it is not sensitive enough.

We thank the reviewer for pointing this issue, as this information is very valuable to understand where the test may has flaws and help readers to decide wether to use this method or not.

Line 150 “Moreover, the enzymes that cleaved the dyes when the cell was in a cell death pathway but they were synthesised at the time the cell was viable [44]”is not understandable. Rephrase it.

We hope the phrase is more comprehensible now. We wanted to note that, enzymes can be synthesized while the cell is viable (e.g., at t=0h) and remain present after the cell has died or is undergoing cell death pathways (e.g., at t=20h).

Section in line numbers 193-200: You disscuss about use of crystal violet as fluorescent dye but it is actually  an alkaline dye that binds to DNA in the nucleus of a cell. It is not fluorescent. Example of fluorescent dye that binds DNA and reflects the number of cell divisions is CFSE which should be mentioned in this part of the text instead of crystal violet.

We appreciate the reviewer's observation that crystal violet (CV) is not fluorescent. The OECD classifies CV in the same group as Sulforhodamine B because both attach to proteins and DNA, even though CV is not fluorescent. Additionally, we thank the reviewer for reminding us about CFSE, which we have included in the corresponding OECD category (lipid-soluble dyes cleaved by esterases within the cell).

Line 197 for crysal violet you state “so this method measures the population mass rather than individual cells”. It measures the number of adherent cells.

We agree with the reviewer and acknowledge the inadequate explanation. We added crystal violet with Sulforhodamine B , as given by the OECD. Our intention was to highlight that crystal violet penetrates adherent cells, which are presumably viable. The amount of dye is then measured with a spectrophotometer and correlated to the number of viable (attached) cells. However, several factors, unrelated to cell viability (e.g., osmolarity, concentration, loss of membrane integrity), can influence the amount of dye that penetrates the cell.

Line 215” Moreover, some test items interfere with the assay e.g., by reducing the dye because interference testing is recommended [62)” Provide examples of MTT reduction such as reduction in specific cell mediums, or substances such as polyphenols etc.

We provided two examples: one where formazan production is reduced and another where it is enhanced. This approach helps the reader better understand the test's limitations.

Lines 264,265 rephrase in one better structured sentence“Rapid and high-throughput test. The test is not invasive, and the cells are kept intact, which enables monitoring the cells for longer periods”

We have addressed the issue as recommended by the reviewer.

Line 266 How do you mean it measures cell mass?

We kindly ask the reviewer to state a specific reference from the 266 because we can't find any refernce to cell mass measurement in line 266.

Line 269 to 271 The statements are not clear. What is the point in this text?

We appreciate the reviewer pointing out that this sentence was nonsensical and misplaced in the paragraph. We have removed it.

Figure 3 is missing

We apologize for the confusion. We initially intended to include a graph to summarize the decision process. However, we decided to describe it in detail instead and deleted the figure, but mistakenly left the caption. We added the figure anyways for a more graphical view.

Line 365 change the term anti-cancerogenic

We have addressed the issue as recommended by the reviewer.

Line 369-377 This is not a good description of the instruments used nor the approaches they offer for assessing cell viability. Rewrite this section.

We have addressed the issue as recommended by the reviewer.

Section 377 to 378 is not clear, Rewrite it.

We have addressed the issue as recommended by the reviewer.

Round 2

Reviewer 1 Report

Comments and Suggestions for Authors

The manuscript has been significantly improved, and I acknowledge that much work has been put into it.

Still, the reversibility of apoptosis must mentioned with at least one reference, e.g. (Tang et al., 2017 Molecular signature of anastasis for reversal of apoptosis 2017 doi: 10.12688/f1000research.10568.2), likewise, cellular senescence was elegantly introduced in v.2 with an appropriate reference (Moseniak and Sikora 2010).

After this small revision and the attentive correction of minor misprints, the article can be recommended for publication. 

Author Response

The manuscript has been significantly improved, and I acknowledge that much work has been put into it.

Still, the reversibility of apoptosis must mentioned with at least one reference, e.g. (Tang et al., 2017 Molecular signature of anastasis for reversal of apoptosis 2017 doi: 10.12688/f1000research.10568.2), likewise, cellular senescence was elegantly introduced in v.2 with an appropriate reference (Moseniak and Sikora 2010).

We agree with the reviewer that this important process was missing, as it was one of the reasons that motivated us to develop new viability methods. We have added the reference in line 170 because anastasis must be especially considered when selecting the target molecules of the apoptotic pathway.

After this small revision and the attentive correction of minor misprints, the article can be recommended for publication.

Reviewer 2 Report

Comments and Suggestions for Authors

The authors have addresed most of the reviewers concerns.

There are few minor issues that should be corrected prior to publishing:

Line 95 delete “the trademark of"

 Line 109” Azqueta et al. noted that the LDH method is not very sensitive for determining cytotoxicity”

You still haven’t explained to my previous question, in what way it is not sensitive enough. What is the reason behind lack of sensitivity in Azqueta’s paper?

Line 185 Figure legend 3 ?” This measurement allows for the quantification of cell mass.” Also in line 204 “population mass”

How do you mean it measures cell mass? It is not correct.

Line 226 use expression “universal ” rather than “sharp”

Line 224. Add dot at the end of sentence

Line 262 This method IS (missing I)

Line 278 determineD (missing D)

Line 281 Correct english :” These methods focus on the direct measurement of cell viability, adhering to the cell death definition established by the NCCD in 2015, which considers a cell dead when its membrane loses integrity.”

Line 299 correct the english: “This step enhances the accuracy of cell viability assessments, as doubled cells are also depolarized for relatively long periods of time.”

Line 301 influenceD (add D)

Line 320 put IS instead of it's

Line 322 TO be addressed

Line detailed (add ED)

Line 327 provide the name of the Table 1

Line 333 correct “best suits better their” to better suites their

Line 362 encompass(omit ED)

Comments on the Quality of English Language

The manuscript should be carefully proofread for typing mistakes and grammar.

Author Response

The authors have addresed most of the reviewers concerns.

There are few minor issues that should be corrected prior to publishing:

Line 95 delete “the trademark of"

We have addressed the issue as recommended by the reviewer.

 Line 109” Azqueta et al. noted that the LDH method is not very sensitive for determining cytotoxicity”

You still haven’t explained to my previous question, in what way it is not sensitive enough. What is the reason behind lack of sensitivity in Azqueta’s paper?

We apologize for overlooking the issue raised by the reviewer in the previous round. We agree with the reviewer that the problem was not appropriately described. We have now included a more detailed description of the LDH sensitivity issue highlighted by Azqueta et al. in their paper. We believe the revised text now more accurately addresses the LDH method issue, highlighting inter-laboratory inconsistencies when measuring cell viability using the LDH method.

Line 185 Figure legend 3 ?” This measurement allows for the quantification of cell mass.” Also in line 204 “population mass”

How do you mean it measures cell mass? It is not correct.

We agree with the reviewer that it does not measure cell mass. We have adopted the OECD terminology, which categorizes such methods as those that measure cell mass. We have corrected the terminology to accurately reflect that the method measures the protein content in the cell.

Line 226 use expression “universal ” rather than “sharp”

We have addressed the issue as recommended by the reviewer.

Line 224. Add dot at the end of sentence

We have addressed the issue as recommended by the reviewer.

Line 262 This method IS (missing I)

We have addressed the issue as recommended by the reviewer.

Line 278 determineD (missing D)

We have addressed the issue as recommended by the reviewer.

Line 281 Correct english :” These methods focus on the direct measurement of cell viability, adhering to the cell death definition established by the NCCD in 2015, which considers a cell dead when its membrane loses integrity.”

We have addressed the issue.

Line 299 correct the english: “This step enhances the accuracy of cell viability assessments, as doubled cells are also depolarized for relatively long periods of time.”

We have addressed the issue.

Line 301 influenceD (add D)

We have addressed the issue as recommended by the reviewer.

Line 320 put IS instead of it's

We have addressed the issue as recommended by the reviewer.

Line 322 TO be addressed

We have addressed the issue as recommended by the reviewer.

Line detailed (add ED)

We have addressed the issue as recommended by the reviewer.

Line 327 provide the name of the Table 1

We have addressed the issue.

Line 333 correct “best suits better their” to better suites their

We have addressed the issue as recommended by the reviewer.

Line 362 encompass(omit ED)

We have addressed the issue as recommended by the reviewer.